A novel approach to secure communication in mega events through Arabic text steganography utilizing invisible Unicode characters

Khan Esam Ali eakhan@uqu.edu.sa
The Custodian of the Two Holy Mosques Institute for Hajj and Umrah Research, Umm Al-Qura University , Makkah , Saudi Arabia
Sperlì Giancarlo
Electronic publication date: 2024 Aug 15
Publication date: 2024
Volume: 10
Electronic Location ID: e2236
Received 2024 Feb 28; Accepted 2024 Jul 13
Copyright: ©2024 Khan
Copyright year: 2024
Copyright holder: Khan
License: This is an open access article distributed under the terms of the Creative Commons Attribution License, which permits unrestricted use, distribution, reproduction and adaptation in any medium and for any purpose provided that it is properly attributed. For attribution, the original author(s), title, publication source (PeerJ Computer Science) and either DOI or URL of the article must be cited.
License URL: https://creativecommons.org/licenses/by/4.0/

Keywords: Arabic text steganography, Invisible unicode characters, Mega events

Funding: Umm Al-Qura University University Rector decree 4502002286 The authors received no funding for this work. Umm Al-Qura University supported this research through a paid sabbatical leave year (2023 –2024) by the University Rector decree No. (4502002286). The funders had no role in study design, data collection and analysis, decision to publish, or preparation of the manuscript.

==============================
Mega events attract mega crowds, and many data exchange transactions are involved among organizers, stakeholders, and individuals, which increase the risk of covert eavesdropping. Data hiding is essential for safeguarding the security, confidentiality, and integrity of information during mega events. It plays a vital role in reducing cyber risks and ensuring the seamless execution of these extensive gatherings. In this paper, a steganographic approach suitable for mega events communication is proposed. The proposed method utilizes the characteristics of Arabic letters and invisible Unicode characters to hide secret data, where each Arabic letter can hide two secret bits. The secret messages hidden using the proposed technique can be exchanged via emails, text messages, and social media, as these are the main communication channels in mega events. The proposed technique demonstrated notable performance with a high-capacity ratio averaging 178% and a perfect imperceptibility ratio of 100%, outperforming most of the previous work. In addition, it proves a performance of security comparable to previous approaches, with an average ratio of 72%. Furthermore, it is better in robustness than all related work, with a robustness against 70% of the possible attacks.

Introduction

The internet has completely changed the technological means via which data is shared in many social interactions. As globalization has progressed, there has been a higher dependence on social media to share data by both individuals and organizations especially during mega events (Al-Khaldy et al., 2022). When a text message is sent via short message service (SMS), email, or social media, the information included in the message is transmitted as plain text, which implies that such information is vulnerable to malicious attacks and unauthorized access. In some cases, this information may be sensitive or confidential, such as passwords, banking credentials, or even some secrets that an individual seeks to send to a friend. As a result, there is an increasing need for intelligence and multimedia security studies that incorporate covert communication, whose primary goal is data concealment (Ahvanooey et al., 2019).

Text hiding, also known as information or data hiding, has gathered a lot of interest recently because of its widespread use and its applications in the network communication and cybersecurity sectors. It is the process of embedding secret data such that it is invisible to adversary or casual readers using a cover text. In the context of mega events, data hiding is important for several reasons. Mega events attract massive crowds, making it easier for eavesdroppers to blend in and covertly listen to conversations. The sheer volume of people creates a challenging environment for security personnel to monitor every individual effectively. Mega events often involve the exchange of sensitive information among organizers, security teams, and other stakeholders. Eavesdropping on these communications can lead to the compromise of security protocols, emergency plans, or confidential details. Mega data breaches, which usually involve millions of compromised data records, can incur significant costs. Figure 1 shows the cost of mega data breaches in 2023 (Invisible Characters, 2022). As depicted in the figure, the greater the amount of data compromised, the higher the cost. As the number of cyber attacks are increasing dramatically in recent mega events, as depicted in Fig. 2 (Poitevin, 2023; SOCRadar, 2022), the cost of data breaches due to these attacks could be substantial unless data is protected in a secure manner. Data hiding helps protect this information from unauthorized access or interception by potential threats. Furthermore, mega events typically use tickets or access credentials for attendees. Data hiding can be employed to embed additional security features, making it more difficult for counterfeiters to replicate tickets or access passes. This enhances the overall security of the event by reducing the risk of unauthorized entry. In addition, security teams and organizers need to communicate discreetly to ensure the effectiveness of security measures. Data hiding allows for covert communication, reducing the risk of intercepted messages and maintaining the element of surprise in security operations. Also, attendees at mega events may have concerns about privacy, especially in the age of pervasive surveillance. Data hiding techniques can be used to secure personal information, making it more challenging for malicious actors to exploit or misuse attendees’ data (Ahvanooey et al., 2019).

Figure 1 The cost of mega breaches in 2023 as reported in (IBMSecurity, 2023).

Figure 2 Number of cyber attacks in some mega events.

Information hiding is classified into two categories in practice: watermarking and steganography. Watermarking safeguards cover media against damaging attacks such as modifications, forgeries, and plagiarism by demonstrating ownership. In contrast, steganography is concerned with the invisible transmission of confidential information in such a way that no one can detect it. In other words, the goal of steganography is concealing the fact that a medium contains secret data (Ahvanooey et al., 2019).

Steganography is the art and science that involves hiding messages within other messages to conceal information. The phrase “secret message” refers to the concealed message, while “cover message” or “cover media” refers to the message used to cover the secret message. The resulting message, which combines the cover message and the secret message, is known as the “stego message” (Souvik, Banerjee & Sanyal, 2011; Gutub, Al-Alwani & Mahfoodh, 2010). The goal of steganography is to conceal the secret message by making use of any redundancy in the cover message while maintaining the integrity of the cover media (Gutub & Fattani, 2007).

Basically, any digital file format can be used as a cover media. Based on the cover media, steganography techniques can be classified into five main categories, image, audio, video, text, and network steganography (Jebur et al., 2023). Information is concealed using the noise present in the cover media in image, audio, and video steganography. On the other hand, in text steganography, redundancy to conceal secret information can take on different forms, including altering a text’s formatting, changing some words, and using random character sequences (Bennett, 2004). In network steganography, data is concealed by making use of the header and payload fields of network protocols. This is achieved by creating hidden channels between a secret sender and a secret recipient (Jebur et al., 2023).

Text-based steganography poses unique challenges due to the scarcity of redundant information in text files compared to other carrier files like images or audio, making it a particularly challenging form of steganography to implement effectively. However, text steganography offers several practical advantages. It requires minimal memory for storage, facilitating efficient memory usage. Its lightweight nature enables seamless transmission over networks, ensuring swift data exchange. Additionally, it reduces printing costs by embedding data within textual content, making it cost-effective for printing. Text steganography is also widely used across social media platforms for covert communication due to its simplicity and long-standing history of use. These practical benefits of text steganography emphasize its efficiency, versatility, and adaptability across various communication channels and mediums, making it the most suitable tool for secure communication in mega events. Therefore, this article focuses on text steganography as the preferred method for ensuring discreet and confidential communication amidst the unique challenges of mega events.

More specifically, this work focuses on the Arabic language, which holds considerable importance as a widely spoken language across numerous countries. It holds profound cultural and religious significance as the language of the Quran, the central religious text in Islam. With approximately 1.6 billion Muslims worldwide, Arabic serves as the language of a significant portion of the global population (Al-Nofaie & Gutub, 2020). Moreover, given that one of the largest and most significant mega events in the world, Hajj, occurs within an Arabic-speaking context, targeting the Arabic language in this article is both logical and pertinent.

The proposed method in this article is based on hiding secret bits within Arabic letters through the utilization of invisible Unicode characters, where each letter can hide two secret bits. With this capacity of each letter, this technique offers a notably greater capacity than previous approaches.

The article is organized as follows. ‘Background Material’ provides a background material about the features of Arabic language and Unicode encoding utilized in this study. In ‘Related Work’, a review of the related work is covered. The methodology of the proposed work is explained in ‘Methodology’. In ‘Experimental Results’ the experimental results of the proposed work are discussed and a comparison with similar previous methods are discussed in ‘Discussion and Comparison’. Finally, the article is concluded in ‘Conclusion’ highlighting some potential directions for future research.

Background Material

This work utilizes some characteristics of Arabic letters and the Unicode encoding system. In this section, these features and characteristics are highlighted. First, the characteristics of Arabic letters is discussed in ‘Arabic letter characteristics’. Then, the features of the Unicode encoding is covered in ‘Unicode characteristics’

Arabic letter characteristics

The Arabic language holds a significant position among human languages, and it is one of the sixth official languages of the United Nations. Around 400 million individuals worldwide use Arabic as their primary language. Its alphabet comprises 28 characters. Notably, Arabic letters are typically connected to each other in written texts, unlike languages like English, where characters are usually written individually (Thabit et al., 2021). Arabic language and its characters have some characteristics that might be useful when thinking about data hiding (Al-Nofaie, Fattani & Gutub, 2016; Alanazi, Khan & Gutub, 2022).

The unique right-to-left writing system of Arabic, contrasting with the left-to-right orientation of languages like English or French, is a significant characteristic. This unidirectional feature not only affects letter arrangement but also shapes the structure of written Arabic, including numerical representation. This consistency aids Arabic readers in processing text visually, ensuring a cohesive reading experience. Understanding this aspect is crucial in applications like data hiding and steganography, emphasizing the need to preserve readability while concealing information within Arabic text.

Another characteristic is connectivity of letters to neighbouring letters within words. Letters can occur at the start, end, middle of a word, or isolated. Some Arabic letters connect with both preceding and following letters, while others only join with preceding ones. Additionally, certain letters cannot connect to either preceding or following ones. Table 1 shows the connected and un-connected Arabic letters.

Furthermore, in Arabic script, some letters are characterized by the presence of dots, while others are free of dots, as shown in Table 2. The dots may be one, two or three, and they may be above or beneath the letter.

Table 1 Connected and un-connected Arabic letters.

	
Arabic letters that may be connected to previous or following letters		
Arabic letters that may be connected only to previous letters		
Arabic letter that may not be connected to previous nor following letters		

Table 2 Dotted and un-dotted Arabic letters.

	
Arabic letters that have dots		
Arabic letters that are free of dots		

These characteristics are not unique for Arabic languages. Other languages, such as Persian and Urdu, have the same characteristics. Therefore, the proposed methodology could be adapted to be utilized in such languages.

Unicode characteristics

Unicode, established in 1987, serves as a comprehensive text encoding standard, accommodating characters from diverse languages worldwide. Its integration into Internet protocols, operating systems, and programming languages ensures interoperability, fostering seamless communication across global platforms. Comprising three encoding forms: UTF-8, UTF-16, and UTF-32, Unicode offers versatility to encode characters efficiently. UTF-8, a variable-width encoding form, is well-suited for languages primarily using Latin characters, offering compatibility with ASCII while supporting various scripts. Conversely, UTF-16 employs a fixed-width approach, striking a balance between efficiency and character support, albeit with potentially higher storage requirements. UTF-32, with a fixed width of four bytes per character, ensures consistent encoding length but may result in increased memory usage, particularly for texts dominated by ASCII characters. Each encoding form caters to distinct use cases, with UTF-8 valued for compactness and compatibility, UTF-16 for efficiency and character diversity, and UTF-32 for its uniform encoding length (Ahvanooey et al., 2019; Product Knowledge, R&D, 2022).

Invisible Unicode characters, known as zero-width characters, play a crucial role in text structure and layout despite lacking a visible representation when rendered. True invisible characters, devoid of visible elements, are ideal for steganographic purposes, enabling covert embedding of information without altering text appearance. Another category includes invisible characters introducing added space into text, subtly influencing spacing between words or characters while remaining visually imperceptible. These characters offer flexibility for specific formatting needs or text layout adjustments. Understanding the behaviour and application of invisible Unicode characters is vital, especially in contexts like steganography, where concealing information within text is paramount (Product Knowledge, R&D, 2022; Invisible Characters, 2022).

Related Work

Various approaches have been proposed in Arabic text steganography, utilizing different techniques to conceal information within Arabic text. The main categorizes of these techniques are based on Arabic letters characteristics, diacritical marks, kashida character, Unicode encoding, and Arabic poetry system (Thabit et al., 2021).

One of the approaches that are based on Arabic letters characteristics involves altering the placement or arrangement of dots within Arabic letters to encode hidden information. By strategically manipulating dot patterns, steganographers can embed covert messages while maintaining the appearance of the text. Early research has explored the utilization of points within Arabic and Persian letters to conceal sensitive information. For instance, in Shirali-Shahreza & Shirali-Shahreza (2006) single secret bit (0 or 1) was hidden within Arabic letters by adjusting the positions of the dots. A similar study proposed in Odeh et al. (2012) addressed Arabic letters with multiple points, assigning two bits to each multipoint letter to double the hidden bits. However, a challenge arose with this method: the retyping process, which could destroy concealed bits. The authors proposed a solution by consolidating all data into a single file to limit future font changes. While this approach enhances capacity and decreases suspicion in the covert text, it is characterized by longer processing times, fixed output format, and vulnerability to retyping or scanning.

Another feature of Arabic letters, which is the sharp edges and geometric shapes, was also utilized to hide secret information. In Roslan, Mahmod & Udzir (2011), this technique was employed by using two keys to determine whether dotted or undotted letters are used for concealing secret bits and specifying the precise positions within the letters where the information is hidden. Each edge of a letter is assigned a unique code, facilitating the concealment of bits within those edges. Similarly, in Mersal et al. (2014), the concept of sharp edges was utilized, with a 24-bit random key concealed within the initial sharp edges of the letters in the cover text. The positioning of the letter within the text determines the sequence of random numbers used to extract the binary representation of the secret message. This binary number forms the code sequence for the message, with each code number representing a binary bit concealed within a specific character of the text. Furthermore, Roslan et al. (2014) introduced the primitive structural method, which incorporates sharp edges, dots, and typo proportion from calligraphy writing. Each Arabic character offers multiple hiding spots for secret bits, determined by its structural features and proportion calculation, enhancing the capacity for concealing information within the text.

Diacritical marks, such as vowel signs and other phonetic symbols, have been also utilized to encode hidden information within the text, exploiting the subtle variations in their placement and appearance. In Aabed et al. (2007), it was noted that the fatha diacritic is almost as prevalent as other diacritics in Arabic, leading to the suggestion of using it to encode “1”, with other diacritics representing “0”. However, this approach may attract undue attention from readers. An improved method proposed in Gutub et al. (2010b) introduced two algorithms based on the number of diacritics needed to conceal secret bits, with one utilizing fixed-size blocks of secret bits and the other mapping consecutive bits of the same value to their respective run lengths. Additionally, Gutub et al. (2008) explored steganography using multiple diacritics, employing both textual and image-based approaches. In Bensaad & Yagoubi (2013), three steganographic methods employing diacritics were discussed: one involving the inclusion or removal of diacritics based on secret bit values, another employing a switch technique, and a third introducing parity bits for each letter. Moreover, Memon & Shah (2011) employed reversed fatha in Arabic and Urdu text to convey concealed messages, a method that matches hidden messages character by character to cover articles. Ahmadoh & Gutub (2015) presented an embedding algorithm utilizing kasrah and fatha diacritics to hide messages, fragmenting messages into binary value arrays and concealing them within respective diacritics based on their parity. In another work, Malalla & Shareef (2016a) introduced a modified fatha for Arabic text steganography, which encrypts secret messages using AES before concealing them within text using a slightly altered fatha to avoid detection. Recently, Gutub (2024) proposed a hiding technique based on multiple diacritics. The author presented two models for the proposed technique. In each model, two diacritics are selected to hide secret bits. If the secret bit is 1, the selected diacritic is kept and the neighbouring diacritics other than the two selected diacritics are deleted. On the other hand, if the secret bit is 0, the selected diacritic is deleted and the neighbouring diacritic is preserved.

In another approach, kashida (also called extension or tatweel character) is utilized for steganography. Kashida, which is a typographical feature used in Arabic script to elongate certain letters for aesthetic or spacing purposes without affecting the content of the written message, can be manipulated to encode hidden data by varying the length or position of the elongated strokes within the text. In Gutub, Al-Alwani & Mahfoodh (2010), kashida was strategically employed with one kashida concealing “0” and two consecutive kashidas hiding “1”, optimizing the process by encoding all possible forms of Arabic letters using 6 bits and introducing a “finishing character” to denote completion. The MSCUKAT algorithm proposed in Al-Nazer & Gutub (2009) scans cover objects to identify suitable letter locations for kashida insertion based on secret bit values. Another algorithm proposed by the authors of Al-Haidari et al. (2009) encodes secret messages as numbers by strategically inserting kashidas within extendable letters. This concept was further developed by the authors of Gutub & Al-Nazer (2010) with the implementation of MSCUKAT. In Gutub et al. (2007), kashida was utilized with both pointed and un-pointed letters to conceal secret bits, while Gutub et al. (2010a) and Al-Haidari et al. (2009) enhanced security by selectively utilizing potential kashida positions and employing secret keys for bit-carrying positions. The kashida variation algorithm (KVA) in Odeh, Elleithy & Faezipour (2013) aimed to enhance robustness by randomly concealing bits in text blocks. Similarly, Alhusban & Alnihoud (2017) proposed four embedding strategies to conceal two secret bits using kashida after specific letters. Techniques such as compressing secret messages using Gzip and encrypting with AES, as shown in Malalla & Shareef (2016b), have been integrated with kashida embedding methods, along with other approaches like hiding voice files within text files using kashida and specific words (Al-Oun & Alnihoud, 2017). Despite the effectiveness of kashida for steganography, challenges such as susceptibility to suspicion and large output file sizes persist. Moreover, efforts to streamline algorithm complexity for improved extraction of hidden information remain limited.

In literature, Unicode characters have also been utilized for concealing information within text. One of the oldest proposals for using white spaces to hide secret bits was presented in Bender et al. (1996), where three scenarios were proposed to hide bits within text, the first at the end of sentences, the second at the end of lines, and the third between words. In Shirali-Shahreza & Shirali-Shahreza (2010), the resemblance between certain Arabic and Persian letters, such as “ ” and “ ” has been exploited, where Persian letters are used to represent “0” and Arabic letters for “1”. Additionally, in Shirali-Shahreza & Shirali-Shahreza (2008b), the isolated and connected forms of Arabic and Persian letters are leveraged to encode binary data, with the ZWJ character aiding in rendering the isolated forms. Furthermore, zero-space characters like ZWJ and ZWNJ were used in Shirali-Shahreza & Shirali-Shahreza (2008d) to hide secret bits by controlling letter connections. The special “La” character in Arabic is employed in Shirali-Shahreza (2007) and Shirali-Shahreza & Shirali-Shahreza (2008a) for concealing secret bits, with different representations used to encode “0” and “1”. Moreover, each Arabic letter’s multiple Unicode codes were utilized in Shirali-Shahreza & Shirali-Shahreza (2008c) to hide secret bits, distinguishing between representative and positional codes for encoding binary data. The concealment is performed word by word, as combining representative and contextual codes within a single word isn’t feasible due to text viewer limitations. The approach introduced in Alanazi, Khan & Gutub (2022) resolved this issue by employing pseudo-spaces and extension characters, thereby enhancing the method’s capacity for concealing information. In Obeidat (2017), three scenarios for concealing information in Arabic text were explored, involving character alterations, preserving cover letters, and adjusting Unicode based on letter type. Two Arabic text steganography methods were proposed in Al-Nofaie, Gutub & Al-Ghamdi (2021). The first, called Kashida-PS, builds upon previous work in Gutub, Al-Alwani & Mahfoodh (2010) and Al-Nofaie, Fattani & Gutub (2016) by incorporating the Kashida feature with PS. The second method, called PS-betWords, utilizes PS between words. Furthermore, Alotaibi & Elrefaei (2018) presented two Arabic text watermarking methods leveraging word spaces, with the first method enhancing a previous approach proposed in Alotaibi & Elrefaei (2016) by incorporating dotting features, and the second method selectively inserting different spaces based on watermark bits.

The Arabic poetry system, as described in Khan (2014), is capable of hiding secret bits, as each Arabic poem contains a representation of binary units. The approach involves assuming that the positions of embedded binary bits within poems contain the secret bits. The actual secret bit corresponds either to the binary position or its reverse. To enhance the capacity of this embedding technique, diacritics and kashida approaches are employed.

The combination of multiple steganographic techniques can enhance the concealment and robustness of hidden information. For example, combining Unicode with kashida was employed in several methods. In Al-Nofaie, Fattani & Gutub (2016), kashida is inserted between Arabic letters to represent a secret bit of one, while white spaces are used to represent zero, with two consecutive white spaces indicating the presence of a zero. A similar approach was adopted in Taha, Hammad & Selim (2020), where kashida and Unicode methods utilize small spaces for concealing bits, with specific patterns indicating the presence of a one. Moreover, Malalla & Shareef (2017) employed techniques like Gzip compression and AES encryption alongside kashida and Unicode methods to embed secret messages. In Alanazi, Khan & Gutub (2022), Medium Mathematical Spaces (MSPs), ZWJ, ZWJN, and kashida were combined to conceal one secret bit per alteration of format, whitespace, or kashida insertion. Kadhem & Ali (2017) proposed a method that merges Unicode with diacritics, utilizing RNA encoding for secret messages and non-printed characters to obscure codes, with compression using modified run length encoding (RLE). Lastly, Alshahrani & Weir (2017) explored the integration of kashida with diacritics, where fatah and consecutive kashidas are used to conceal different bits, demonstrating the versatility of combining different encoding techniques for steganographic purposes.

Table 3 summarizes the main pros and cons of previous work, classified by the techniques used. From this table, it is evident that methods utilizing kashida or diacritics raise suspicion, as kashida are not commonly used in most Arabic texts, and diacritics, when used, should appear on all letters, not just some. Other techniques, such as those using the points of letters or similar letters, have low capacity because not all letters can be utilized to hide secret data. Although the technique of using sharp edges does not add visible characters and can hide more secret bits, it is not robust enough due to the need for a reference table and code sequence that must be securely transmitted to the receiver and correctly retrieved. In general, methods utilizing Unicode have better imperceptibility, as they minimize the addition of visible characters. However, most of these methods cannot hide more than one bit per character and some suffer from incorrect connectivity between letters due to the use of different Unicode forms. Table 4 provides a more detailed analysis of methods based on Unicode techniques.

Table 3 Summary of the related work and their main characteristics.

Ref.	Techniques employed	pros	cons	
Shirali-Shahreza & Shirali-Shahreza (2006)	Points of letters	Robustness against insertion and deletion attacks, and also retyping and copying attacks if stego text is converted to image	High computational time
Low capacity
Data hidden may be lost if stego text is retyped or copied and pasted.	
Odeh et al. (2012)	
Roslan, Mahmod & Udzir (2011)	Sharp edges	No visible characters are added, which increases imperceptibility.
Hiding secret bits in sharp edges increases capacity.	The reference tables or code sequence must be sent with the stego text in a secure manner.
Data hidden may be lost if stego text is retyped or copied and pasted.	
Mersal et al. (2014)	
Roslan et al. (2014)	
Aabed et al. (2007)	Diacritics	Robustness against retyping and copying attacks as long as diacritics are not added or deleted.	Using diacritics in some letters and not in others raises suspicions.
Data hidden may be altered by adding or deleting some diacritics.	
Gutub et al. (2010b)	
Gutub et al. (2008)	
Bensaad & Yagoubi (2013)	
Memon & Shah (2011)	
Ahmadoh & Gutub (2015)	
Malalla & Shareef (2016a)	
Gutub (2024)	
Gutub, Al-Alwani & Mahfoodh (2010)	Kashida	Robustness against copying attacks.	Not all letters accept kashida, which decreases capacity.
Data hidden may be altered by adding or deleting some kashidas.
Having kashida in some locations of the text raises suspicions.	
Gutub et al. (2007)	
Al-Nazer & Gutub (2009)	
Al-Haidari et al. (2009)	
Gutub & Al-Nazer (2010)	
Gutub et al. (2010a)	
Al-Haidari et al. (2009)	
Odeh, Elleithy & Faezipour (2013)	
Alhusban & Alnihoud (2017)	
Malalla & Shareef (2016b)	
Al-Oun & Alnihoud (2017)	
Alshahrani & Weir (2017)	Kashida + diacritics	Using more than one method to hide bits increases capacity.	Having kashida and diacritics in some locations of the text raises suspicions.	
Khan (2014)	Arabic poetry + kashida + diacritics	All letters can hide secret bits.	Having kashida and diacritics in some locations of the text raises suspicions.
Limited to Windows-1256 encoding	
Shirali-Shahreza & Shirali-Shahreza (2010)	Similar letters with different codes	No extra characters are added.	Only some letters are used to hide secret bits, which decreases capacity.	
Shirali-Shahreza (2007)	
Shirali-Shahreza & Shirali-Shahreza (2008a)	
Shirali-Shahreza & Shirali-Shahreza (2008d)	Unicode	Visible characters added are minimized which improves imperceptibility.
Extra characters added are minimal.	In most techniques, not all characters are utilized to hide secret bits, which degrades capacity.
In some cases, characters are not connected properly, which degrades imperceptibility.	
Bender et al. (1996)	
Shirali-Shahreza & Shirali-Shahreza (2008c)	
Shirali-Shahreza & Shirali-Shahreza (2008b)	
Obeidat (2017)	
Kadhem & Ali (2017)	
Al-Nofaie, Gutub & Al-Ghamdi (2021)-Method 2	
Alotaibi & Elrefaei (2018)-Method 2	
Alotaibi & Elrefaei (2016)	Unicode + dotted letters	No visible characters are added, which increases imperceptibility.	Bits are hidden word by word, which decreases capacity.	
Alotaibi & Elrefaei (2018)-Method 1	
Alanazi, Khan & Gutub (2022)	Unicode + kashida	Extra characters added are minimal.
Improved capacity over kashida-based methods.	Kashidas added are visible, which degrades imperceptibility.	
Al-Nofaie, Gutub & Al-Ghamdi (2021)-Method 1	
Al-Nofaie, Fattani & Gutub (2016)	
Taha, Hammad & Selim (2020)	
Malalla & Shareef (2017)	

Table 4 A detailed review of the related work based on Unicode.

Ref.	Techniques employed	Methodology	Pros	Cons	
M1	Bender et al. (1996)	Unicode	Normal spaces are inserted to hide secret bits at the end of sentences, at the end of lines, or between words. If hidden between words, every secret bit is hidden using two spaces, and an extra space is inserted for encoding	Hiding of secret bits does not depend on characters of cover text, so it can be implemented in any language	Bits are hidden word by word, which lowers capacity.
Extra spaces degrade imperceptibility	
M2	Alotaibi & Elrefaei (2016)	Unicode + dotted letters	ZWNJ is used to hide secret bits. it is inserted or not based on the last letter of the word whether dotted or not.	No visible characters are added, which increases imperceptibility.	Bits are hidden word by word, which decreases capacity.	
M3	Alotaibi & Elrefaei (2018) -1	Unicode + dotted letters	ZWNJ is used to hide secret bits. it is inserted or not based on the last letter of the word before it and the first letter of the word after it, whether dotted or not.	No visible characters are added, which increases imperceptibility.	Bits are hidden word by word, which decreases capacity.	
M4	Alotaibi & Elrefaei (2018) -2	Unicode	Four white spaces (PS, HS, TS, and ZWS) are used to hide 4 bits after each word.	Hiding of secret bits does not depend on characters of cover text, so it can be implemented in any language.	Some white spaces are visible, which decreases imperceptibility.
ZWS (200B) cannot be sent over Gmail.	
M5	Al-Nofaie, Fattani & Gutub (2016)	Unicode + kashida	Kashida between letters and whitespaces between words are inserted to hide secret bits.	Improved capacity over kashida-based methods.	Not all letters are utilized to hide secret bits, which decreases capacity.
Kashidas added are visible, which degrades imperceptibility.	
M6	Al-Nofaie, Gutub & Al-Ghamdi (2021)-1	Unicode + kashida	Kashida is used between two connected letters, and PS is inserted between two unconnected letters, and also after normal spaces.	All letters can be utilized to hide secret bits.	Kashidas added are visible, which degrades imperceptibility.	
M7	Al-Nofaie, Gutub & Al-Ghamdi (2021)-2	Unicode	PS’s are inserted between words to hide secret bits	No visible characters are added, which increases imperceptibility.
Hiding of secret bits does not depend on characters of cover text, so it can be implemented in any language	The number of PS’s to be added may be high for some secret bit sequences.	
M8	Shirali-Shahreza & Shirali-Shahreza (2008d)	Unicode	One bit is hidden in each letter. If it is connected to the next letter,
ZWJ is inserted for hiding bit 1 and not inserted for hiding bit 0. If the letter is not connected to the next letter, ZWNJ is used instead.	No visible characters are added, which increases imperceptibility	Not all characters are utilized to hide secret bits.	
M9	(Taha, Hammad & Selim, 2020)	Unicode + kashida	Kashida is inserted when possible to hide 1, and then three white spaces (Thin space, Hair space, and Six-PRE-EM space) are used between words to hide 3 secret bits.	Improved capacity over kashida-based methods.	Kashidas and some white spaces are visible, which decreases imperceptibility.	
M10	Shirali-Shahreza & Shirali-Shahreza (2008c)	Unicode	Representative and positional codes of Parsian and Arabic letters are used to hide secret bits word by word.	No extra characters are added.	Bits are hidden word by word, which lowers capacity.
In some cases, characters are not connected properly, which degrades imperceptibility.	
M11	Shirali-Shahreza & Shirali-Shahreza (2008b)	Unicode	Representative and positional codes of Parsian and Arabic letters are used to hide secret bits in each letter. ZWJ is inserted if representative and positional codes come after each other.	All letters can be utilized to hide secret bits.
Extra characters added are minimal.
No visible characters are added.	In some cases, characters are not connected properly, which degrades imperceptibility.	
M12	Alanazi, Khan & Gutub (2022)	Unicode + kashida	General and contextual codes of Arabic letters are used to hide secret bits. kashida, ZWJ, and ZWNJ are used for connecting letters when different codes come after each other.	Extra characters added are minimal.	Using kashida decreases imperceptibility.
In some cases, characters are not connected properly, which degrades imperceptibility.	
M13	Malalla & Shareef (2017)	Unicode + kashida	To hide 1, kashida is used when possible, or the code of Arabic letter is changed from isolated to medium or vice versa.	Extra characters added are minimal.	Using kashida decreases imperceptibility.
In some cases, characters are not connected properly.	
M14	Kadhem & Ali (2017)	Unicode	Some non-printed ASCII characters are used to hide 1, along with isolated and connected forms of Unicode	All characters are utilized to hide secret bits which increases capacity.	The non-printed ASCII characters are only invisible in some platforms. In others, they are shown	
M15	Obeidat (2017)	Unicode	Representative and positional codes of letters not joining with previous character are used to hide secret bits	Extra characters added are minimal.	Not all letters are utilized to hide secret bits, which decreases capacity.
In some cases, characters are not connected properly, which degrades imperceptibility.	

The proposed method in this study aims to leverage the benefits of Unicode techniques while addressing the limitations of previous work.

Methodology

The methodology of the proposed technique is based on hiding secret bits on the letters of a cover text utilizing invisible Unicode characters. The utilization of invisible Unicode characters to hide secret data has been proposed in several previous works, as discussed in ‘Related Work’. However, the novelty of the proposed methodology in this work is based on combining the utilization of invisible Unicode characters with two characteristics of the Arabic letters explained in ‘Arabic Letter Characteristics’. The first is whether the letter is dotted or not, which has been utilized for hiding secret data in several previous works such as (Roslan, Mahmod & Udzir, 2011; Alhusban & Alnihoud, 2017) and (Alotaibi & Elrefaei, 2018). The second is whether the letter is connected or not, which has not been utilized before –up to our knowledge –as proposed in this work.

Based on the two characteristics of each Arabic letter of a word, a two-bit value DC is assigned to that letter, as shown in Table 5. The two-bit DC of each character of the word is XORed with two secret bits. The result of the XOR is a two-bit value EN. Based on EN, an invisible white space is appended at the end of the word. Using this methodology, each letter hides two secret bits and will correspond to one white space. Therefore, four invisible characters are required.

Table 5 The two-bit value given for each Arabic character.

	Dotted	Undotted	
Connected	1 1	0 1	
Not connected	1 0	0 1	

The selection of invisible Unicode characters to be utilized in the proposed method is based on the following criteria:

1. Any invisible Unicode character inserted should not be visible in any way. In other words, the cover Arabic text should not be visually changed after inserting the invisible Unicode character. This increases the imperceptibility of the proposed method.

2. The inserted invisible Unicode characters should not be omitted when the stego text is sent via communication media or copied and pasted through different software. This increases the robustness of the proposed method.

Among the available invisible Unicode characters, we selected the four invisible characters shown in Table 6, which all satisfy the above mentioned criteria (Invisible Characters, 2022; Unicode Explorer, 2024).

In addition to the above four invisible white spaces used to hide secret bits, two other invisible characters are used at the end of the hiding process, as will be explained below. These two characters are \ufe00 and \ufe01.

UTF-16 is used in this work because it has fixed size of two bytes, while UTF-8 has different sizes. Specifically, some of the Arabic letters and the selected invisible characters are represented in more bytes using UTF-8 than UTF-16.

The advantages of the proposed method compared to previous work can be summarized in the following:

1. The proposed method can hide two secret bits for each character, while all previous methods based on Unicode could hide one secret bit per character at most. This advantage increases the capacity of the proposed method in comparison to other works.

2. Using the four selected white spaces to hide secret bits after words of the cover text increases the imperceptibility of the stego text. In other words, no one can notice the insertion of these white spaces in the stego text since there is no visible difference between the cover text and the stego text.

3. Adding one white space for each character helps in detecting alterations of the stego text, since the number of white spaces after each word should be equal to the number of characters in that word.

In addition, it is worth noting that secret bits could be hidden directly using the selected four invisible Unicode characters instead of XORing them with the characteristics of the corresponding Arabic letter. However, using the proposed XORing step has several advantages:

Table 6 Invisible Unicode characters that satisfy the first criterion.

Unicode character	Unicode number	Description	
Right-to-left mark (RLM)	\u200f	Used in computerized typesetting for bi-directional text, affecting the grouping of adjacent characters according to text direction.	
Arabic letter mark (ALM)	\u061c	Utilized in computerized typesetting for bi-directional text, similar to the Right-to-left mark (RLM), but with some differences in its effects on bidirectional level resolutions for nearby characters.	
Zero width non-joiner (ZWNJ)	\u200c	Causes two characters that would normally be joined into a ligature to be printed in their final and initial forms, respectively.	
Combining grapheme joiner (CGJ)	\u034f	Used to semantically separate characters that should not form digraphs and to prevent canonical reordering of combining marks during text normalization.	

1. The secret bits cannot be extracted directly from the Unicode characters without knowing the characteristics of the corresponding letter in the text. This is important if someone has noticed the insertion of invisible characters and tried to extract secret bits from them.

2. The XORing with the characteristics of the Arabic letters will scramble the pattern of secret bits. The advantage of this is specifically important for structured secret bits, such as all ones, all zeros, or alternating patterns.

The proposed method consists of two main processes: one for embedding the secret message in the cover text, and the other for extracting it from the stego text. In the following, these two processes are explained in detail.

Embedding algorithm

The embedding process, which is shown in detail in Algorithm 1 , consists of the following steps:

1. Preprocessing: In this step, the number of secret bits is checked. If it is odd, a redundant “0” is appended to the end of the secret bits. This is because each character will hide two secret bits, so the number of secret bits should be even. In addition, the number of characters in the cover text should be more than or equal to twice the number of secret bits.

2. Hiding secret bits: For each word of the cover text, the following steps are performed:

(a) Find the number of characters in each word, which is checked by reading characters until a normal space (\u0020) is found.

(b) For each character of the word, a two-bit value DC is given based on its characteristics, as shown in Table 5.

(c) The two-bit DC of each character of the word is XORed with two secret bits S 1S 0. The result of the XOR is a two-bit value EN; as illustrated in the following equation: (1) EN=DC⊕S1S0

(d) Based on EN, one of the invisible white spaces is appended at the end of the word, as shown in Table 7.

(e) The above steps are repeated until all secret bits are hidden.

3. Post-processing: After all secret bits are encoded, an invisible Unicode character is appended to indicate the number of secret bits, whether odd or even. For odd number of secret bits, \ufe00 is inserted. Otherwise, \ufe01 is appended.

A flowchart of the embedding algorithm is shown in Fig. 3, which highlights the main steps of the algorithm. Table 8 shows an example of the embedding process, where the inserted invisible white spaces are bolded.

Table 7 The corresponding invisible white space for the two-bit value EN.

EN	The corresponding invisible white space	
0 0	\u061c	
0 1	\u200f	
1 0	\u200c	
1 1	\u034f	

Figure 3 Flowchart of the embedding process.

	
Algorithm 1: Embedding Process	
Input: Secret bits, Cover text	
Output: Stego text	
1.	i= 1	
2.	CharCount= 0	
3.	SecretCount= number of secret bits	
4.	ifSecretCount mod 2 = 1	
5.	Append 0 to secret bits \\add a redundant bit if number of secret bits is odd	
6.	end if	
7.	while not End of Secret Bits	
8.	Read next two secret bits S 1S 0	
9.	Read ith character Text[i ] of the cover text	
10.	T [i ] = Unicode(Text[i ])	
11.	whileT [i ] ! = space (\u0020)	
12.	CharCount++ \\count number of characters in each word	
13.	ifT [i ] is dotted	
14.	D [i ] = 1	
15.	else	
16.	D [i ] = 0	
17.	end if	
18.	ifT [i ] is not connected	
19.	C [i ] = 0	
20.	else	
21.	C [i ] = 1	
22.	end if	
23.	DC [i ] =D [i ] & C [i ] \\& is append operation	
24.	EN=DC [i ] ⊕S 1S 0	
25.	caseEN:	
26.	00: W [CharCount ] = \u061c	
27.	01: W [CharCount ] = \u200f	
28.	10: W [CharCount ] = \u200c	
29.	11: W [CharCount ] = \u034f	
30.	end case	
31.	i ++	
32.	Read Text[i]	
33.	T[i] = Unicode(Text[i])	
34.	end while	
35.	j=i –1	
36.	for (count= 1 to CharCount; count ++)	
37.	insert W [count ] at T [j ]	
38.	end for	
39.	CharCount= 0	
40.	end while	
41.	ifSecretCount mod 2 = 0	
42.	T [i +1] = \ufe01 \\indicates end of even number of secret bits	
43.	else	
44.	T [i +1] = \ufe00 \\indicates end of odd number of secret bits	
45.	end if	
46.	for (k= 1 to i +1, k ++)	
47.	OutputFile[k ] = Character(T [k ]) \\convert unicode into characters	
48.	end for	
49.	while not End of cover text \\copy the remaining characters from cover text to output file	
50.	Read ith character Text[i] of the cover text	
51.	OutputFile[k ] = Text[i ]	
52.	k++; i++	
53.	end while	
		

Extracting algorithm

Algorithm 2 shows the details of the extracting process, which is depicted in Fig. 4. The algorithm consists of the following steps:

Table 8 An example of the embedding process.

The bolding indicatines where invisible white spaces are inserted.

	
Cover text		
Unicode of the cover text	\u0644\u0627\u0020 \u0625\u0644\u0647\u0020 \u0625\u0644\u0627\u0020 \u0627\u0644\u0644\u0647\u0020 \u0645\u062d\u0645\u062f\u0020 \u0631\u0633\u0648\u0644\u0020 \u0627\u0644\u0644\u0647	
Secret bits	0100011111100011111011	
Unicode of the Stego text	\u0644\u0627\u061c\u061c\u0020 \u0625\u0644\u0647\u200f\u200c\u200c\u0020 \u0625\u0644\u0627\u200c\u200f\u034f\u0020 \u0627\u0644\u0644\u0647\u034f\u034f\u200c\ufe01\u0020 \u0645\u062d\u0645\u062f\u0020 \u0631\u0633\u0648\u0644\u0020 \u0627\u0644\u0644\u0647	

Figure 4 Flowchart of the extracting process.

1. Read a character from the stego text until one of the two invisible white spaces indicating the end of secret bits (\ufe00 or \ufe01) is found.

2. Until the read character is a space (\u0020), do the following:

(a) Count the number of Arabic letters and the number of invisible white spaces.

(b) For each Arabic character, find the corresponding two-bit value DC, as explained in Table 5.

3. For each word, the value of each Arabic Letter DC is XORed with the value EN of the corresponding invisible white space, as shown in Table 7, which results in two of the secret bits S 1S 0. This is illustrated using the following equation: (2) S1S0=DC⊕EN

4. If the last read character is the invisible white spaces (\ufe00), which indicates an odd number of secret bits, ignore the last extracted secret bit.

	
Algorithm 2: Extracting Process	
Input: Stego text	
Output: Extracted secret bits	
1.	i= 1	
2.	CharCount= 0	
3.	SpaceCount= 0	
4.	Read ith character Text[i ] of the Stego text	
5.	T [i ] = Unicode(Text[i ])	
6.	whileT [i ] ! = (\ufe00 or \ufe01) \\not end of secret bits	
7.	whileT [i ] ! = (\u0020) \\not end of a word	
8.	caseT [i ]:	
9.	\u061c:	
10.	SpaceCount ++	
11.	S [SpaceCount ] = 00	
12.	\u200f:	
13.	SpaceCount ++	
14.	S [SpaceCount ] = 01	
15.	\u200c:	
16.	SpaceCount ++	
17.	S [SpaceCount ] = 10	
18.	\u034f:	
19.	SpaceCount ++	
20.	S [SpaceCount ] = 11	
21.	others:	
22.	CharCount ++	
23.	ifT [i ] is dotted	
24.	D [CharCount ] = 1	
25.	else	
26.	D [CharCount ] = 0	
27.	end if	
28.	ifT [i ] is not connected	
29.	C [CharCount ] = 0	
30.	else	
31.	C [CharCount ] = 1	
32.	end if	
33.	DC [CharCount ] =D [CharCount ] & C [CharCount ]	
34.	end case	
35.	i ++	
36.	Read Text[i ]	
37.	T [i ] = Unicode(Text[i ])	
38.	end while	
39.	ifCharCount ! =SpaceCount	
	print(“Text may be altered.”)	
	else	
	forj= 1 to SpaceCount; j ++	
40.	S 1S 0=DC [j ] ⊕S [j ]	
41.	Insert S 1S0to OutputFile	
42.	end for	
	CharCount= 0	
43.	SpaceCount= 0	
44.	i ++	
45.	Read Text[i ]	
46.	T[i ] = Unicode(Text[i ])	
	end if	
47.	end while	
48.	ifT [i ] = \ufe00     \\odd number of secret bits	
49.	Delete last bit from OutputFile	
50.	end if	
		

Experimental Results

The proposed method has been implemented using the Python 3 programming language. In order to evaluate our proposed steganography method, performance measurements of steganography techniques should be applied. Steganography techniques are assessed based on four key criteria: capacity, imperceptibility, robustness, and security. In the following, the experimental results of the proposed method in terms of these four measurements are discussed.

Capacity

Capacity refers to the amount of secret data that can be hidden within a cover object (Alotaibi & Elrefaei, 2018). Various methods have been applied in the literature to measure the capacity of steganographic techniques (Khan, 2014). In this work, the capacity is calculated using the following equation: (3) Capacity ratio=C×SCH×100

where C is the number of characters capable of hiding secret bits, S is the number of secret bits per character, and CH is the total number of characters in the cover text.

Using this equation, the capacity ratio of the proposed method can be calculated as follows: (4) Capacity ratio=CH−SP×SCH×100

where SP is the number of normal spaces in the cover text.

It should be noted that, in order to deal with texts including all kinds of characters, such as diacritics and connecting characters, the developed algorithm is designed with the assumption that any character in the text other than Arabic letters is treated as an undotted and connected letter.

The experimental results of the proposed method are presented in Table 9. The cover texts selected for testing are the last 30 Surahs of the Holy Quran, both with and without diacritics. We used four types of secret messages: all ones, all zeros, alternating ones and zeros, and random ones and zeros. The table displays the statistics of the 30 Surahs and the results of hiding the secret messages within them. These results include the maximum number of secret bits that a cover can hide and the capacity ratio. It should be noted that all types of secret messages produced the same results, demonstrating that the performance of the proposed method does not depend on the structure of the secret messages.

Table 9 Capacity ratio of the proposed method.

The bolded values are the capacity and security ratios, which are the important results shown in the tables.

	surah number in Quran	surah name	Number of letters	Number of diacritics	Number of spaces	Number of words	Results without diacritics	Results with diacritics	
							Total number of characters	maximum # of secret bits	Capacity ratio	Total number of characters	maximum # of secret bits	Capacity ratio	
1	114	An-Nas	80	65	19	20	99	160	162%	164	290	177%	
2	113	Al-Falaq	73	67	22	23	95	146	154%	162	280	173%	
3	112	Al-Ikhlas	47	42	14	15	61	94	154%	103	178	173%	
4	111	Al-Masad	81	68	22	23	103	162	157%	171	298	174%	
5	110	An-Nasr	80	68	18	19	98	160	163%	166	296	178%	
6	109	Al-Kafirun	99	78	26	27	125	198	158%	203	354	174%	
7	108	Al-Kawthar	43	41	9	10	52	86	165%	93	168	181%	
8	107	Al-Ma‘un	114	95	24	25	138	228	165%	233	418	179%	
9	106	Quraysh	77	64	16	17	93	154	166%	157	282	180%	
10	105	Al-Fil	97	87	22	23	119	194	163%	206	368	179%	
11	104	Al-Humazah	134	122	32	33	166	268	161%	288	512	178%	
12	103	Al-‘Asr	73	56	13	14	86	146	170%	142	258	182%	
13	102	At-Takathur	123	112	27	28	150	246	164%	262	470	179%	
14	101	Al-Qari‘ah	160	126	35	36	195	320	164%	321	572	178%	
15	100	Al-‘Adiyat	169	145	39	40	208	338	163%	353	628	178%	
16	99	Az-Zalzalah	158	138	35	36	193	316	164%	331	592	179%	
17	98	Al-Bayyinah	404	334	93	94	497	808	163%	831	1,476	178%	
18	97	Al-Qadr	115	102	29	30	144	230	160%	246	434	176%	
19	96	Al-‘Alaq	288	252	71	72	359	576	160%	611	1,080	177%	
20	95	At-Tin	162	128	33	34	195	324	166%	323	580	180%	
21	94	Ash-Sharh	102	94	26	27	128	204	159%	222	392	177%	
22	93	Ad-Duhaa	165	141	39	40	204	330	162%	345	612	177%	
23	92	Al-Layl	314	272	70	71	384	628	164%	656	1,172	179%	
24	91	Ash-Shams	253	201	53	54	306	506	165%	507	908	179%	
25	90	Al-Balad	342	287	81	82	423	684	162%	710	1,258	177%	
26	89	Al-Fajr	586	494	138	139	724	1,172	162%	1,218	2,160	177%	
27	88	Al-Ghashiyah	382	327	91	92	473	764	162%	800	1,418	177%	
28	87	Al-’A’la	296	251	71	72	367	592	161%	618	1,094	177%	
29	86	At-Tariq	254	214	60	61	314	508	162%	528	936	177%	
30	85	Al-Buruj	469	374	108	109	577	938	163%	951	1,686	177%	

Security and imperceptibility

Imperceptibility measures how well the technique preserves the quality and appearance of the cover object, ensuring that any changes are indiscernible to human observers, while security assesses the difficulty of unauthorized parties detecting or extracting the hidden message without knowledge of the embedding method or key, ensuring the confidentiality and integrity of the hidden data (Alotaibi & Elrefaei, 2018). Security and imperceptibility are closely related as the aim of both is to make it difficult to detect the presence of hidden data within the cover text.

For imperceptibility, the proposed method does not add any extra visible characters beyond those originally present in the cover text, thus achieving a very high imperceptibility with 100% imperceptibility ratio.

For security evaluation, the security ratio is computed using the following equation (Al-Nofaie, Gutub & Al-Ghamdi, 2021): (5) Security ratio=Amount after discountOriginal character×100

where the amount after discount is calculated using the following equations: (6) Discount value=Original characters×Excess characters100

(7) Amount after discount=Original characters-Discount value

To evaluate the performance of the proposed method, we conducted a number of experiments using the last third Surah of the Holy Quran, Surah ‘Al-Ikhlas’, as the cover text. The results are summarized in Table 10 and illustrated in Fig. 5. In these experiments, different sizes of secret messages were hidden in the cover text, both with and without diacritics. As depicted in the figure, the security of the proposed method increases as the number of secret bits decreases, which shows that there is a tradeoff between capacity and security, as proved in Zhang & Li (2004).

Table 10 Security ratio of the proposed method.

The bolded values are the capacity and security ratios, which are the important results shown in the tables.

Results without diacritics	Results with diacritics	
# of secret bits	Excess characters	Discount value	Amount after discount	Security ratio	# of secret bits	Excess characters	Discount value	Amount after discount	Security ratio	
94	50	30.5	30.5	50%	178	92	94.76	8.24	8%	
47	25	15.25	45.8	75%	89	46	47.38	55.62	54%	
24	13	7.93	53.1	87%	44	23	23.69	79.31	77%	
12	7	4.27	56.7	93%	2	12	12.36	90.64	88%	

Figure 5 Security ratio of the proposed method.

Robustness

Robustness measures the ability of hidden data to remain intact during transmission and to endure various attacks or modifications to the cover object, while still enabling reliable extraction of the hidden message. The key aspects of robustness include resistance to changes during transmission, resilience against attacks or modifications, and the reliable extraction of hidden data.

For the first aspect, and since the proposed method is intended for communication in mega events, it is crucial to test it across various platforms and software to verify that the stego file retains all hidden data without loss and that all secret bits can be extracted accurately. To prove this, three main tests were conducted on a sample cover text:

1. The first test involved running the algorithm and opening the output stego file using different software applications such as MS Word and Windows Notepad. The stego text was then employed to extract the secret message.

2. The second test consisted of sending the stego text via email using various email platforms such as Gmail and Outlook. The received message was then used to extract the secret bits.

3. The third test entailed sending the stego text via WhatsApp and subsequently extracting the secret bits from the sent message.

In all the aforementioned tests, the integrity of the stego texts was confirmed, and the proposed method successfully extracted the correct message.

To evaluate the proposed method against possible attacks, which involve deliberate modifications or interventions to disrupt, remove, or manipulate the hidden information within a stego-object (Alotaibi & Elrefaei, 2018), the following experiments were conducted:

1. Localized insertion: where some noise is inserted in one location of the stego text.

2. Dispersed insertion: where some noise is inserted in different locations of the stego text.

In these tests, adding noise in the form of characters, words, or sentences without adding white spaces was detected in the extraction process. This is because each character should correspond to a white space.

3. Localized deletion: where a random text is deleted from one location of the stego text.

4. Dispersed deletion: where random texts are deleted from different locations of the stego text.

These tests include deleting characters, words, or entire sentences from the stego text. In most cases, this deletion was detected during the extraction process. The cases that were not detected occurred when a whole word or sentence was deleted from the middle of the text. This is because deleting a whole word or sentence removes both characters and white spaces. However, the word or sentence to be deleted must be chosen carefully to keep the remaining text meaningful, which makes this type of attack more difficult.

5. Copying and pasting: where the stego text was copied and pasted in another program or file.

6. Formatting: where the style of stego text was changed, such as: font style, text size, coloring, highlighting and any other effects.

In these tests, the secret bits were extracted correctly because copying and pasting of the stego text will not remove the invisible characters even if pasted in different software, and formatting the stego text has no effect on the invisible characters.

7. Retyping: where the stego text was retyped.

8. Printing: where the stego text was scanned and read through OCR software and then printed.

Because invisible characters are lost when the stego text is retyped or scanned or read through OCR and reprinted, the proposed method is not robust against these types of attacks.

Table 11 summarizes the robustness of the proposed method against these types of attacks.

Table 11 Analysis of the robustness of the proposed method against possible attacks.

Robustness attack	Robust?	
Localized insertion (character, word, sentence)	√	
Dispersed insertion (characters, words, sentences)	√	
Localized deletion of a character	√	
Localized deletion of a word from the middle of the stego text	×	
Localized deletion of a word from the end of the stego text	√	
Localized deletion of a sentence from the middle of the stego text	×	
Localized deletion of a sentence from the end of the stego text	√	
Dispersed deletion of a character	√	
Dispersed deletion of words from the middle of the stego text	×	
Dispersed deletion of words from the end of the stego text	√	
Dispersed deletion of sentences from the middle of the stego text	×	
Dispersed deletion of sentences from the end of the stego text	√	
Copying and pasting	√	
Formatting	√	
Retyping	×	
Printing	×	

Discussion and Comparison

In this section, comparisons of the performance of the proposed method with related work are discussed. The related methods shown in Table 4 are selected, which all use Unicode in their methodologies. The comparison is discussed in the main measurements of steganography: capacity, security, and robustness.

Capacity

Using Eq. (3) given in ‘Capacity’, the maximum capacity ratio of the related works has been calculated for the last 30 Surahs of the Holy Quran, and the results are shown in Tables 12 and 13. The tables show the capacity ratio when the 30 cover texts include diacritics and when they are free of diacritics, respectively. The capacity ratio and its average in the two cases are illustrated in Figs. 6 and 7.

Table 12 Capacity comparison when cover texts include diacritics.

The bold values show the average of capacity (with and without diacritics).

	M1	M2	M3	M4	M5	M6	M7	M8	M9	M10	M11	M12	M13	M14	M15	Proposed	
1	6%	12%	23%	46%	31%	60%	46%	49%	54%	12%	49%	60%	49%	100%	29%	177%	
2	7%	14%	27%	54%	33%	59%	54%	45%	60%	14%	45%	59%	45%	100%	25%	173%	
3	7%	14%	27%	54%	36%	59%	54%	46%	63%	15%	46%	59%	46%	100%	23%	173%	
4	6%	13%	26%	51%	37%	60%	51%	47%	63%	13%	47%	60%	47%	100%	23%	174%	
5	5%	11%	22%	43%	32%	59%	43%	48%	54%	11%	48%	59%	48%	100%	27%	178%	
6	6%	13%	26%	51%	33%	62%	51%	49%	59%	13%	49%	62%	49%	100%	27%	174%	
7	5%	10%	19%	39%	31%	56%	39%	46%	51%	11%	46%	56%	46%	100%	24%	181%	
8	5%	10%	21%	41%	33%	59%	41%	49%	54%	11%	49%	59%	49%	100%	25%	179%	
9	5%	10%	20%	41%	32%	59%	41%	49%	53%	11%	49%	59%	49%	100%	25%	180%	
10	5%	11%	21%	43%	38%	58%	43%	47%	59%	11%	47%	58%	47%	100%	20%	179%	
11	6%	11%	22%	44%	33%	58%	44%	47%	55%	11%	47%	58%	47%	100%	24%	178%	
12	5%	9%	18%	37%	31%	61%	37%	51%	49%	10%	51%	61%	51%	100%	28%	182%	
13	5%	10%	21%	41%	37%	57%	41%	47%	57%	11%	47%	57%	47%	100%	19%	179%	
14	5%	11%	22%	44%	34%	61%	44%	50%	55%	11%	50%	61%	50%	100%	27%	178%	
15	6%	11%	22%	44%	35%	59%	44%	48%	58%	11%	48%	59%	48%	100%	23%	178%	
16	5%	11%	21%	42%	31%	58%	42%	48%	52%	11%	48%	58%	48%	100%	27%	179%	
17	6%	11%	22%	45%	35%	60%	45%	49%	58%	11%	49%	60%	49%	100%	24%	178%	
18	6%	12%	24%	47%	34%	59%	47%	47%	57%	12%	47%	59%	47%	100%	24%	176%	
19	6%	12%	23%	46%	32%	59%	46%	47%	55%	12%	47%	59%	47%	100%	25%	177%	
20	5%	10%	20%	41%	35%	60%	41%	50%	56%	11%	50%	60%	50%	100%	24%	180%	
21	6%	12%	23%	47%	31%	58%	47%	46%	55%	12%	46%	58%	46%	100%	27%	177%	
22	6%	11%	23%	45%	33%	59%	45%	48%	56%	12%	48%	59%	48%	100%	24%	177%	
23	5%	11%	21%	43%	33%	59%	43%	48%	54%	11%	48%	59%	48%	100%	24%	179%	
24	5%	10%	21%	42%	34%	60%	42%	50%	55%	11%	50%	60%	50%	100%	26%	179%	
25	6%	11%	23%	46%	35%	60%	46%	48%	58%	12%	48%	60%	48%	100%	24%	177%	
26	6%	11%	23%	45%	33%	59%	45%	48%	56%	11%	48%	59%	48%	100%	25%	177%	
27	6%	11%	23%	46%	35%	59%	46%	48%	58%	12%	48%	59%	48%	100%	23%	177%	
28	6%	11%	23%	46%	34%	59%	46%	48%	57%	12%	48%	59%	48%	100%	24%	177%	
29	6%	11%	23%	45%	33%	59%	45%	48%	55%	12%	48%	59%	48%	100%	26%	177%	
30	6%	11%	23%	45%	34%	61%	45%	49%	57%	11%	49%	61%	49%	100%	26%	177%	
Average	6%	11%	22%	45%	34%	59%	45%	48%	56%	12%	48%	59%	48%	100%	25%	178%	

Table 13 Capacity comparison when cover texts are free of diacritics.

The bold values show the average of capacity (with and without diacritics).

	M1	M2	M3	M4	M5	M6	M7	M8	M9	M10	M11	M12	M13	M14	M15	Proposed	
1	10%	19%	38%	77%	52%	100%	77%	81%	90%	20%	81%	100%	81%	166%	48%	162%	
2	12%	23%	46%	93%	57%	100%	93%	77%	103%	24%	77%	100%	77%	171%	43%	154%	
3	11%	23%	46%	92%	61%	100%	92%	77%	107%	25%	77%	100%	77%	169%	39%	154%	
4	11%	21%	43%	85%	61%	100%	85%	79%	104%	22%	79%	100%	79%	166%	39%	157%	
5	9%	18%	37%	73%	54%	100%	73%	82%	91%	19%	82%	100%	82%	169%	46%	163%	
6	10%	21%	42%	83%	54%	100%	83%	79%	95%	22%	79%	100%	79%	162%	43%	158%	
7	9%	17%	35%	69%	56%	100%	69%	83%	90%	19%	83%	100%	83%	179%	42%	165%	
8	9%	17%	35%	70%	57%	100%	70%	83%	91%	18%	83%	100%	83%	169%	42%	165%	
9	9%	17%	34%	69%	55%	100%	69%	83%	89%	18%	83%	100%	83%	169%	42%	166%	
10	9%	18%	37%	74%	66%	100%	74%	82%	103%	19%	82%	100%	82%	173%	34%	163%	
11	10%	19%	39%	77%	57%	100%	77%	81%	96%	20%	81%	100%	81%	173%	41%	161%	
12	8%	15%	30%	60%	51%	100%	60%	85%	81%	16%	85%	100%	85%	165%	47%	170%	
13	9%	18%	36%	72%	64%	100%	72%	82%	100%	19%	82%	100%	82%	175%	34%	164%	
14	9%	18%	36%	72%	55%	100%	72%	82%	91%	18%	82%	100%	82%	165%	45%	164%	
15	9%	19%	38%	75%	60%	100%	75%	81%	98%	19%	81%	100%	81%	170%	39%	163%	
16	9%	18%	36%	73%	52%	100%	73%	82%	89%	19%	82%	100%	82%	172%	46%	164%	
17	9%	19%	37%	75%	59%	100%	75%	81%	96%	19%	81%	100%	81%	167%	40%	163%	
18	10%	20%	40%	81%	58%	100%	81%	80%	98%	21%	80%	100%	80%	171%	41%	160%	
19	10%	20%	40%	79%	55%	100%	79%	80%	94%	20%	80%	100%	80%	170%	43%	160%	
20	8%	17%	34%	68%	58%	100%	68%	83%	92%	17%	83%	100%	83%	166%	40%	166%	
21	10%	20%	41%	81%	54%	100%	81%	80%	95%	21%	80%	100%	80%	173%	46%	159%	
22	10%	19%	38%	76%	56%	100%	76%	81%	94%	20%	81%	100%	81%	169%	41%	162%	
23	9%	18%	36%	73%	56%	100%	73%	82%	93%	18%	82%	100%	82%	171%	41%	164%	
24	9%	17%	35%	69%	56%	100%	69%	83%	91%	18%	83%	100%	83%	166%	43%	165%	
25	10%	19%	38%	77%	58%	100%	77%	81%	97%	19%	81%	100%	81%	168%	41%	162%	
26	10%	19%	38%	76%	56%	100%	76%	81%	94%	19%	81%	100%	81%	168%	42%	162%	
27	10%	19%	38%	77%	60%	100%	77%	81%	98%	19%	81%	100%	81%	169%	38%	162%	
28	10%	19%	39%	77%	57%	100%	77%	81%	95%	20%	81%	100%	81%	168%	41%	161%	
29	10%	19%	38%	76%	55%	100%	76%	81%	93%	19%	81%	100%	81%	168%	44%	162%	
30	9%	19%	37%	75%	56%	100%	75%	81%	94%	19%	81%	100%	81%	165%	43%	163%	
Average	9%	19%	38%	76%	57%	100%	76%	81%	95%	20%	81%	100%	81%	169%	42%	162%	

As depicted in the figures, our method outperforms all prior approaches across all cover texts when diacritics are included in the cover texts with an average capacity ratio of 178%. The main feature of our proposed method that gives this superior capacity ratio is that, in our method, two secret bits are hidden in each character and that all characters except the normal spaces are utilized for hiding secret bits. In some of the other methods only one bit per character is hidden at most, while in some others, the hidden data depends on the number of spaces, which is always less than the number of characters. Furthermore, diacritics are not utilized in hiding secret bits in most of the previous approaches.

For the case when the cover texts are free of diacritics, only M14 has slightly better capacity ratio than our method. The main reason for this is that M14 utilizes normal spaces to hide secret bits, while our method does not.

Security and imperceptibility

As shown in Table 4, one of the main drawbacks of the methods based on the use of different Unicode forms is that they suffer from incorrect connectivity between some letters, even though they tried to avoid that. For instance, the example given in Alanazi, Khan & Gutub (2022) shows that the sentence “ ” has changed to “ ” after hiding the secret message. It is clear that the letter “ ” does not connect correctly to the previous letter. Another example is given in Obeidat (2017), where also the letter “ ” does not connect correctly to the previous letter. This degrades the imperceptibility of these methods. Therefore, in this section, these methods, which are M10 to M15, are excluded from the comparison. In addition, the approaches that have very low capacity have also been excluded from the comparison, because the secret message will be very small if these approaches are included. These approaches are the ones that hide secret messages word by word, which are M1 to M3.

To evaluate the performance of the proposed method against the other related approaches in terms of security and imperceptibility, we conducted five experiments using five different secret messages: all ones (Exp1), all zeros (Exp2), alternating ones and zeros (Exp3), random ones and zeros with more ones than zeros (Exp4), and random ones and zeros with more zeros than ones (Exp5). The cover text selected is the last third Surah of the Holy Quran, Surah ‘Al-Ikhlas’. The number of bits in the secret messages is the maximum that could be hidden among all methods, which is 56 bits. The results of these experiments are shown in Table 14 and depicted in Fig. 8.

Figure 6 Capacity comparison with related work when cover texts include diacritics.

Figure 7 Capacity comparison with related work when cover texts are free of diacritics.

Table 14 Security and imperceptibility comparison.

The bolded texts show the average of security and imperceptibility.

	method	M4	M5	M6	M7	M8	M9	Proposed	
Exp1	imperceptibility ratio	54%	8%	2%	100%	100%	8%	100%	
Security ratio	44%	44%	-12%	-68%	44%	44%	72%	
Exp2	imperceptibility ratio	100%	100%	51%	100%	100%	100%	100%	
Security ratio	100%	100%	44%	100%	100%	100%	72%	
Exp3	imperceptibility ratio	77%	54%	28%	100%	100%	54%	100%	
Security ratio	72%	72%	16%	52%	72%	72%	72%	
Exp4	imperceptibility ratio	74%	48%	25%	100%	100%	48%	100%	
Security ratio	68%	68%	12%	34%	68%	68%	72%	
Exp5	imperceptibility ratio	82%	62%	31%	100%	100%	62%	100%	
Security ratio	77%	77%	21%	44%	77%	77%	72%	
average imperceptibility ratio		77%	54%	27%	100%	100%	54%	100%	
average security ratio		72%	72%	16%	32%	72%	72%	72%	

As illustrated in the figure, the proposed method, along with M7 and M8, achieves the highest imperceptibility ratio of 100%.

Regarding security, as discussed in ‘Security and imperceptibility’, the proposed method maintains the same security ratio regardless of the secret bits. In contrast, the security of other approaches varies based on the structure of the secret bits. As illustrated in the figure, the proposed method exhibits a commendable level of security, with a 72% security ratio on average, outperforming M6 and M7, and achieving equal average security with the other approaches, namely, M4, M5, M8, and M9.

Figure 8 Security and imperceptibility comparison.

Robustness

The robustness against the possible attacks discussed in ‘Robustness’ has been analyzed for the related works in comparison with our proposed method. This analysis is shown in Table 15. The analysis shows that all methods are not robust against the retyping attack, because the inserted extra characters may not be retyped. In addition, all methods are not robust against deletion of words and sentences, because the extra characters used to hide secret bits will be lost. For the deletion of a character, it can only be detected by the proposed method, along with M4 and M7, whereas the other approaches are not robust against such an attack. This is because M4 and M7 hide secret bits by inserting white spaces after a word, regardless of the characters of that word, while the hiding of bits in the other approaches depends on the characters of the text, so deleting a character will change the secret message. For our approach, deleting a character without the corresponding white space will be detected in the extracting process.

Furthermore, only M1 and M9 are robust against printing attacks because they rely on visible characters for hiding secret bits, while the others utilize invisible characters that will be lost when the stego text is printed. On the other hand, all methods are robust against copying and pasting and formatting, because the inserted extra characters are not affected or lost when the text is copied or formatted.

For the insertion attacks, only M4, M7, M10, and the proposed method are robust against insertion of a character. For M4 and M7, adding a character will not affect the inserted white spaces, so the secret message does not change. For M10, if the character added is of the same code, that will not affect the secret message, and if it is different, then that will be detected. In our prosed method, inserting a character without a corresponding white space will be detected in the extracting process. For all other approaches, adding a character will affect the secret message.

However, the only method that is robust against insertion of words or sentences is our method, because these added words or sentences will not correspond to white spaces, and that will be detected. For the other approaches, adding a whole word is not detected because it will be treated as having secret bits.

In addition, only our proposed approach is robust against insertion or deletion of words or sentences at the end of the stego text, because the ending white space will not be available, which will be detected in the extraction process.

Table 15 Robustness comparison.

Attack		M1	M2	M3	M4	M5	M6	M7	M8	M9	M10	M11	M12	M13	M14	M15	Proposed	
Localized insertion	character	x	x	x	√	x	x	√	x	x	√	x	x	x	x	x	√	
word	x	x	x	x	x	x	x	x	x	x	x	x	x	x	x	√	
sentence	x	x	x	x	x	x	x	x	x	x	x	x	x	x	x	√	
Dispersed insertion	character	x	x	x	√	x	x	√	x	x	√	x	x	x	x	x	√	
word	x	x	x	x	x	x	x	x	x	x	x	x	x	x	x	√	
sentence	x	x	x	x	x	x	x	x	x	x	x	x	x	x	x	√	
Localized deletion	character	x	x	x	√	x	x	√	x	x	x	x	x	x	x	x	√	
word (middle)	x	x	x	x	x	x	x	x	x	x	x	x	x	x	x	x	
sentence (middle)	x	x	x	x	x	x	x	x	x	x	x	x	x	x	x	x	
word (end)	x	x	x	x	x	x	x	x	x	x	x	x	x	x	x	√	
sentence (end)	x	x	x	x	x	x	x	x	x	x	x	x	x	x	x	√	
Dispersed deletion	character	x	x	x	√	x	x	√	x	x	x	x	x	x	x	x	√	
word (middle)	x	x	x	x	x	x	x	x	x	x	x	x	x	x	x	x	
sentence (middle)	x	x	x	x	x	x	x	x	x	x	x	x	x	x	x	x	
word (end)	x	x	x	x	x	x	x	x	x	x	x	x	x	x	x	√	
sentence (end)	x	x	x	x	x	x	x	x	x	x	x	x	x	x	x	√	
Copying and pasting	√	√	√	√	√	√	√	√	√	√	√	√	√	√	√	√	
Formatting	√	√	√	√	√	√	√	√	√	√	√	√	√	√	√	√	
Retyping	x	x	x	x	x	x	x	x	x	x	x	x	x	x	x	x	
Printing	√	x	x	x	x	x	x	x	√	x	x	x	x	x	x	x	

To give this comparison a value, we use the following equation to evaluate the robustness of a method: (8) Robustness=number of tests demonstrating the method’s robustnesstotal number of tests.

Using this equation, Fig. 9 illustrates the robustness of the proposed method compared to related approaches. The figure clearly shows that the proposed method is better than all other methods in robustness, with a robustness of 70% against possible attacks.

Conclusion

This article introduces a novel approach to secure communication during mega events using Arabic text steganography with invisible Unicode characters. In the proposed method each Arabic letter can hide two secret bits based on the two characteristics of that letter: whether it is dotted or not, and whether it is connected or not.

The method offers several advantages: it enables the transmission of small secret messages within Arabic text, making it more compact compared to audio or video covers. Moreover, it’s adaptable to social media platforms, which are prevalent communication channels during mega events. Additionally, the stego text can be sent via email or WhatsApp while preserving the embedded messages’ integrity. Given the widespread use of Arabic in communication, particularly during mega events like Hajj and Umrah, this method holds significant relevance.

The proposed method demonstrates superior performance across key steganography metrics. It surpasses previous methods with an average capacity ratio of 178% because it hides two secret bits in all characters of the cover texts, except for normal spaces, whereas the other approaches hide one bit per character at most, and some of them hide less than that. In addition, the proposed method achieves a perfect imperceptibility ratio of 100% because it does not add any visible characters. Furthermore, it exhibits commendable levels of robustness, as it can detect 70% of possible attacks.

Figure 9 Robustness comparison.

Although the proposed method achieved a good level of security of an average of 72% security ratio, with a comparable performance to other related approaches, this aspect is one of its main limitations because the size of the stego text is larger than the cover text. However, since there is a tradeoff between security and capacity, this limitation is acceptable, especially given the perfect imperceptibility of the proposed method.

As a future direction, steganographic techniques with even higher degrees of security and resilience may be developed by utilizing the remarkable imperceptibility and high capacity of the proposed method.

Supplemental Information

Supplemental Information 1 Embedded Code

Supplemental Information 2 Extracting Code

Additional Information and Declarations

Competing Interests

Author Contributions

Data Availability

The authors declare there are no competing interests.

Esam Ali Khan conceived and designed the experiments, performed the experiments, analyzed the data, performed the computation work, prepared figures and/or tables, authored or reviewed drafts of the article, and approved the final draft.

The following information was supplied regarding data availability:

The Python code of the algorithm developed is available in the Supplementary File.

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
