# Peer review of "A novel approach to secure communication in mega events through Arabic text steganography utilizing invisible Unicode characters"

_PeerJ Computer Science, doi:10.7717/peerj-cs.2236_

## Round 0.1 · original submission · Major Revisions

The reviewers underlined the relevance of the manuscript. They suggest to underline the novelties of the proposal as well as more visualization analysis on the evaluation section.

**Language Note:** PeerJ staff have identified that the English language needs to be improved. When you prepare your next revision, please either (i) have a colleague who is proficient in English and familiar with the subject matter review your manuscript, or (ii) contact a professional editing service to review your manuscript. PeerJ can provide language editing services - you can contact us at [email protected] for pricing (be sure to provide your manuscript number and title). – PeerJ Staff

Reviewer 1 ·

Basic reporting

No comment

Experimental design

it would be interesting to carry out the experiment on texts other than Arabic?

Validity of the findings

The article is about A novel approach to secure communication in mega events through Arabic text steganography utilizing invisible Unicode characters. This paper is not acceptable in its current form but has merit. After the corrections presented by the authors, it is suitable for publication in the journal.
The paper should be improved namely in some identified aspects:
1. Is this approach valid for a text other than Arabic? if not why? I believe that this approach must be universal, therefore adapted to any type of text to be validated
2. In figure 1, please correct ‘’capacity compariron’’
3. Authors wrote: The proposed method demonstrates superior performance across key steganography metrics. It surpasses previous methods with an average capacity ratio of 89% and achieves a perfect imperceptibility ratio of 100%.. how can you judge these values are in the accepted level?

Reviewer 2 ·

Basic reporting

1. Enrich the introduction section with visuals, such as diagrams depicting the extent of data compromise across various mega events, to enhance comprehension and engagement or any other that may be relevant.

2. Enhance the literature review or related work section by incorporating a structured comparison table featuring columns detailing authors, publication years, techniques employed, utilized datasets, results obtained, and associated pros and cons.

Experimental design

3. Integrate a visual representation of the methodology, showing key sections and highlighting the unique contributions of the study, to offer a clearer understanding of the research process.

Validity of the findings

4. Conclude the paper by addressing the limitations in the proposed work, thus providing a balanced assessment of its scope and potential constraints.

5. Improve the quality of Figure 2 to ensure its clarity and effectiveness in conveying information.

6. Incorporate a dedicated discussion section within the paper to foster deeper analysis, interpretation, and synthesis of the research findings and their implications.

Reviewer 3 ·

Basic reporting

The methodology lacks details to verify the scientific terms. The authors must:
- Explain the difference of the proposed method with existing methods.
- For some steganography algorithms with key, the encrypt/decrypt flows are verified for the uniqueness of the output. Therefore, the authors must prove the encrypt and decrypt processes also guarantee uniqueness, i.e., there are not two different inputs giving the same output (for encrypt and decrypt too).
- For both processes, the authors also need to add the input/output requirement to the algorithms.
- The math base for the proposed method also needs to be added.

Experimental design

Experimental results are presented insufficiently. The important test for steganography algorithms is the attack test. The authors did not show any theory or relevant statements. They only put a table for the attack test result without any explanation. Other tests such as noise attack, also must be added. For your reference, I recommend the author reference this article: https://doi.org/10.1007/s11042-021-11803-1

Validity of the findings

The discussion should be expanded and improved.
The conclusion also needs to be improved.

Additional comments

- Add some experimental results
- Check typos and grammar mistakes
- Cite some recent related article

---

## Round 0.2 · accepted · Accept

Thank you for submitting the manuscript to PeerJ. I submit acceptance decision on the basis of reviewers suggestions.

Reviewer 1 ·

Basic reporting

No comment

Experimental design

Good

Validity of the findings

After the corrections provided by the authors, it is suitable for publication in the journal.

Reviewer 2 ·

Basic reporting

All changes have been completed

Experimental design

All changes have been completed

Validity of the findings

All changes have been completed

Reviewer 3 ·

Basic reporting

OK

Experimental design

OK

Validity of the findings

OK

Additional comments

No